# Botulinum toxin type A success rates and predictive factors for favourable outcomes in paediatric patients with acquired esotropia

Narisa Rattanalert[1]*, Teechaya Nonboonyawat[1], Supaporn Tengtrisorn[1], Penny Singha[1], Orapan Aryasit[1], Juthamat Witthayaweerasak[1], Manupol Tangthongkum[2]

1 Department of Ophthalmology, Faculty of Medicine, Prince of Songkla University, Hat Yai, Songkhla, Thailand, 2 Department of Otolaryngology Head and Neck Surgery, Faculty of Medicine, Prince of Songkla University, Hat Yai, Songkhla, Thailand

* narisa.r@psu.ac.th

## Abstract

### Purpose

This study aimed to assess the efficacy of botulinum toxin type A in treating childhood-acquired comitant esotropia and identify predictive factors of treatment success.

### Methods

This retrospective, consecutive, non-comparative cohort study included children under 18 years treated with botulinum toxin type A for acquired comitant esotropia between 2013 and 2019, with a minimum 6-month follow-up. The primary outcome was the success rate of achieving horizontal deviation within 10 prism dioptres at 6 months post-botulinum toxin type A treatment.

### Results

In total, 49 children with a mean treatment age $8.10 \pm 4.02$ years were assessed. The three most common types of esotropia were acute acquired comitant, intermittent, and cyclic esotropia (57.2%, 20.4%, and 12.2%, respectively). The mean esodeviation at near and distance was $42.55 \pm 13.39$ and $42.65 \pm 13.35$ prism dioptres, respectively. The botulinum toxin type A treatment success rate was 51% (25/49), with survival analysis indicating a declining cumulative probability of treatment success over time (48%, 28%, and 21% at 12, 18, and 24 months, respectively). Significant predictive factors for successful outcomes were pre-botulinum toxin type A esodeviation >30–50 prism dioptres ($p = 0.008$) and time from diagnosis to treatment <2 months ($p = 0.001$).

**Data availability statement:** The data contain potentially identifying patient information and are subject to restrictions imposed by the Ethics Committee of the Faculty of Medicine, Prince of Songkla University, to protect participant privacy. Qualified researchers who meet the criteria for access to confidential data may request access by contacting the Research Ethics Committee, Faculty of Medicine, Prince of Songkla University (email: medpsu.ec@gmail.com).

**Funding:** Faculty of Medicine, Prince of Songkla University, Hat Yai, Songkhla, Thailand. Funder has no role in the study design, data collection, and analysis, decision to publish, or preparation of the manuscript.

**Competing interests:** The authors have declared that no competing interests exist.

## Conclusion

Botulinum toxin type A demonstrated a success rate of 51% at 6 months with pre-botulinum toxin type A esodeviation and early treatment identified as predictive factors for treatment success suggesting its clinical application.

## Introduction

Esotropia, a form of strabismus, involves an inward deviation of ocular alignment, where one eye turns inward. Childhood-acquired comitant esotropia is commonly seen in paediatric ophthalmology clinics with patients typically seeking medical attention within the first decade of life. Esotropia manifests in accommodative or non-accommodative forms [1,2] and often leads to diplopia, diminished stereopsis and binocularity, or blurred vision in the affected eye, negatively impacting the patient's vision and potentially causing amblyopia. Psychosocially, this condition can affect the patient's self-esteem, interpersonal relationships, and access to vocational opportunities [3]. Treatment options for esotropia include glasses, prisms, occlusion, botulinum toxin, and extraocular muscle surgery, aiming to correct the deviation of the eyes [4].

Surgery has long been the established standard treatment for acquired comitant esotropia, with extraocular muscle surgery widely recognised as the primary approach. Several studies have reported satisfactory outcomes and a success rate of 50–84% with surgery [5–9]. However, an alternative treatment trend has emerged, focusing on the use of botulinum toxin type A (BTXA) for comitant esotropia. Scott AB [10] was the pioneer who introduced the use of BTXA for adult strabismus, paving the way for non-surgical methods in strabismus treatment. Subsequently, the application of BTXA in strabismus, particularly in treating esotropia, has gained wide acceptance with studies confirming its safety and effectiveness [8,11–15]. Additionally, systematic reviews in 2017 showed no significant difference in the effectiveness of botulinum toxin therapy compared with conservative or surgical treatments. They indicated only a slight reduction in the likelihood of achieving correct alignment of the eyes in the botulinum toxin group. However, these trials were noted to have low-certainty evidence due to potential selection bias and methodological limitations [4].

These varied outcomes have raised concerns regarding the effectiveness of BTXA as a treatment for children with acquired comitant esotropia. Therefore, this study ascertained the success rate of BTXA and identify predictive factors associated with successful outcomes using a single BTXA injection in childhood-acquired comitant esotropia.

## Materials and methods

### Patient selection

This retrospective, consecutive, non-comparative cohort study was conducted at Songklanagarind Hospital, Thailand. This study was approved by the Human

Research Ethics Committee (HREC), Faculty of Medicine, Prince of Songkla University (REC.63-008-2-4, Date 2020-03-18), and adhered to the principles of the Declaration of Helsinki. The need for informed consent was waived due to the retrospective and de-identified nature of the data analysed as per the HREC guidelines.

Electronic medical records of children under 18 years of age, diagnosed with acquired comitant esotropia (with an angle of esodeviation not varying by >5 prism dioptres [PD] in all gaze positions or with either eye fix) who underwent BTXA treatment between January 2013 and December 2019, with a follow-up duration of at least 6 months, were reviewed. Acute acquired comitant esotropia was defined as the acute onset of esotropia or diplopia within six months of presentation. Partial accommodative esotropia was diagnosed in patients with hyperopia of +3.00 dioptres (D) or more and a reduction of esotropia by at least 10 PD following hyperopic correction. Intermittent esotropia was defined as esotropia that presented intermittently in patients with hyperopia of less than +3.00 D. Non-refractive accommodative esotropia was characterized by a near deviation exceeding the distance deviation by more than 10 PD. Cyclic esotropia was defined as esotropia occurring in a regular cycle, typically every 24 or 48 hours. Patients with esotropia associated with neurological disorders, developmental abnormalities, prior ocular surgery, or previous strabismus surgery were excluded from study. Electronic medical records were accessed for research purposes between March 20, 2020, and December 30, 2023. All data were de-identified in the recorded form. The authors had access to identifiable information during the data collection phase, but not after data collection was completed.

## Data collection

Baseline characteristics, including sex, age at treatment, symptom duration, types of acquired esotropia, visual acuity, refraction (expressed as spherical equivalent [sphere power + half of cylinder power, dioptres]), presence of diplopia, amblyopia status, time from diagnosis to treatment, and follow-up duration, were recorded. All patients were administered an intramuscular injection of 7.5 units of BTXA (Dysport®) into the medial rectus muscle in both eyes using forceps, without conjunctival incision or electromyographic guidance, by paediatric ophthalmologists in either the operative room under general anaesthesia or outpatient department under topical anaesthesia. Two patients with esotropia of 30 and 35 PD, who received a unilateral injection of 7.5 units of BTXA into the medial rectus muscle, were excluded to ensure more reliable data and to better assess the effectiveness of BTXA treatment. The angle of deviation, stereoacuity, complications, and any additional treatments were recorded during each visit. The primary outcome was the success rate of achieving horizontal deviation within 10 PD at 6 months post-BTXA treatment. Secondary outcomes included binocularity status and identification of predictive factors for successful treatment. Recurrent esotropia was defined as the reappearance of esotropia after an initially successful alignment outcome at 6 months following BTXA treatment.

## Statistical analysis

Data were analysed using the R program version 4.1.0 (The R Group, Vienna, Austria) with the Epicalc software [16]. Descriptive statistics, including frequency, mean ± standard deviation, and median and interquartile range (IQR), were calculated. The Wilcoxon signed-rank test was used to compare pre-BTXA and post-BTXA angles of deviation. The chi-squared test or Fisher's exact test was used to assess categorical variables. Continuous variables were analysed using the Mann–Whitney U test or independent t-test. $p < 0.05$ was considered statistically significant. Univariate and multivariate logistic regression analyses were performed to predict factors associated with successful outcomes. Primary outcome analysis was restricted to the results following the initial BTXA injection. Data from subsequent interventions were presented separately for descriptive purposes and were not included in the primary success rate calculation. The Kaplan–Meier method (Lawless, 1982) was used to calculate the probability of treatment success over time, considering treatment failure or esotropia recurrence as events, and successful follow-up as censoring events.

# Results

## Patients and disease characteristics

Forty-nine patients were included in the study. Baseline characteristics (Table 1) revealed a male-to-female ratio of 2:1. The mean age at treatment initiation was 8.10 ± 4.02 years. The three most prevalent types of esotropia were acute acquired comitant, intermittent, and cyclic esotropia. The mean time from diagnosis to treatment was 4.67 months, with a mean follow-up duration of 23 months. Among the patients, 16.3% presented with pre-treatment amblyopia.

## Success rate of BTXA

The success rate of initial BTXA treatment at 6 months was 51% (25/49) in the overall cohort and 60.7% (17/28) among patients with acute acquired comitant esotropia. Among the 25 patients who achieved successful motor outcomes at 6 months, binocularity data were available for 23. Of these, 22 demonstrated functional binocular vision; all achieved fusion, and 19 achieved stereopsis. The median stereopsis was 40 seconds of arc (IQR: 25–80).

**Table 1. Baseline characteristics.**

| Baseline characteristics | N = 49 (%) or mean ± SD |
|---|---|
| Male sex | 32 (65.3) |
| Age at treatment (years) | 8.10 ± 4.02 |
| Symptoms duration before diagnosis (months) | 3.99 ± 5.43 |
| Time of diagnosis to treatment (months) | 4.67 ± 7.27 |
| **Types of acquired esotropia** | |
| Acute acquired comitant esotropia | 28 (57.2) |
| Intermittent esotropia | 10 (20.4) |
| Cyclic esotropia | 6 (12.2) |
| Partially accommodative esotropia | 3 (6.1) |
| Non-refractive accommodative esotropia | 2 (4.1) |
| **Visual acuity (LogMAR)** | |
| Right eye | 0.18 ± 0.23 |
| Left eye | 0.15 ± 0.14 |
| **Refraction (dioptres)** | |
| Right eye | 0.20 ± 1.89 |
| Left eye | 0.41 ± 1.89 |
| **Amblyopia** | 8 (16.3) |
| Strabismic amblyopia | 6 |
| Combined strabismic and refractive amblyopia | 2 |
| **Diplopia** | |
| Yes | 15 (30.6) |
| No | 20 (40.8) |
| No available data | 14 (28.6) |
| Pre-BTXA esotropia at near (PD) | 42.55 ± 13.39 |
| Pre-BTXA esotropia at distance (PD) | 42.65 ± 13.35 |
| Follow-up duration (months) | 23.02 ± 16.32 |

*BTXA* botulinum toxin type A, *LogMAR* logarithm of the minimum angle of resolution, *PD* prism dioptres, *SD* standard deviation.

In the subgroup with acute acquired comitant esotropia, binocularity data were available for 16 of 17 patients. All 16 achieved binocular function, with a median stereopsis of 40 seconds of arc (IQR: 25–80).

Seven patients in the success group (28%) experienced recurrent esotropia following the initial BTXA injection, with a mean time to recurrence of 14.84±3.55 months. Among them, four patients underwent secondary interventions (one muscle surgery and three additional BTXA treatments), achieving a success rate of 75% for the second intervention.

The mean time to failed treatment was 2.59±1.91 months in the treatment failure group. In this group, 84% (21/25) patients underwent a secondary intervention, specifically, 18 patients underwent muscle surgery and 3 underwent repeat BTXA treatments. The success rates for the secondary interventions were 55.5% (10/18) for muscle surgery and 33.3% (1/3) for BTXA.

Post-BTXA complications occurred in 49% (24/49) patients, including ptosis (n = 10), overcorrection (n = 6), combined ptosis and overcorrection (n = 6), and subconjunctival haemorrhage (n = 2). No further medical or surgical intervention was performed for these complications. Overcorrections resolved over time, within 5–20 weeks.

Comparison of characteristics between the success and failure groups revealed statistically significant differences in pre-BTXA esotropia (>30–50 PD) at near ($p = 0.002$) and distance ($p = 0.016$), time from diagnosis to treatment (<2 months; $p = 0.001$), and follow-up duration ($p = 0.001$; Table 2).

Additionally, the Kaplan–Meier curve depicting disease-free survival illustrated a 51% probability of treatment success at 6 months (95% confidence interval [CI] = 36.4–63.9). However, this probability of success declined to 48% at 12 months (95% CI = 32.9–61.3), 28% at 18 months (95% CI = 14–44.7) and 21% at 24 months (95% CI = 7.5–39.7; Fig 1). The median disease-free survival time was 6 months (2.73–13.23).

### Predictive factors for successful outcome

Predictive factors associated with successful outcomes were analysed in the successful outcomes group at 6 months using univariate and multivariate logistic regression analyses (Table 3). Pre-BTXA near esodeviation ($p = 0.008$) and time from diagnosis to treatment ($p = 0.001$) emerged as significant predictive factors for a successful outcome.

The Kaplan–Meier curve for disease-free survival in patients with pre-BXTA esotropia indicated that >30–50 PD at near and distance significantly influenced successful treatment. The probabilities of success were 67.7% ($p < 0.001$) and 62.9% ($p < 0.001$) at 6 months for near and distance, respectively (Fig 2). Additionally, time from diagnosis to treatment <2 months was correlated with successful outcomes, with a 75% probability of success at 6 months ($p < 0.001$).

### Discussion

Our study provides valuable insights into the success rate of BTXA treatment for acquired comitant esotropia in children and identifies predictive factors for successful outcomes. Overall, we observed a 51% success rate at 6 months post-BTXA. In cases of acute acquired comitant esotropia, the success rate was higher, at 60.7% (17 out of 28 patients). Additionally, pre-treatment esodeviation and time from diagnosis to treatment emerged as significant predictive factors for successful outcomes ($p = 0.008$ and $p = 0.001$, respectively). Acute acquired comitant, intermittent, and cyclic esotropia were the three most prevalent types of acquired esotropia observed in our study.

Although BTXA has been widely used in strabismic treatment, studies examining its long-term success rate and ideal candidates are limited [17,18]. In this regard, our study revealed a 51% success rate with BTXA treatment, showing a decline in success over time. Nevertheless, approximately one-fifth of the patients sustained optimal outcomes beyond 2 years. Our Kaplan–Meier analysis confirmed a progressive decline in the probability of treatment success over time, with only 21% of patients maintaining optimal ocular alignment beyond two years. This trend underscores the importance of long-term follow-up to detect and manage recurrence in a timely manner. Overall, our findings align with previous studies [18], but show varying success rates compared with others. For instance, study by Wan et al. [8] reported higher success rates of BTXA at 81% and 67% at 6 and 18 months, respectively, for acute onset comitant esotropia, whereas our study

**Table 2. Comparison characteristics between the success and failure groups.**

| Characteristics | Success group N = 25 (%) or mean ± SD | Failure group N = 24 (%) or mean ± SD | p-value* |
|---|---|---|---|
| **Sex** | | | |
| Male | 18 (72.0) | 14 (58.3) | 0.315 |
| Female | 7 (28.0) | 10 (41.7) | |
| **Age at treatment (years)** | 8.42 ± 4.53 | 7.77 ± 3.48 | 0.578 |
| **Symptom duration (months)** | 4.07 ± 5.47 | 3.92 ± 5.50 | 0.920 |
| **Types of acquired esotropia** | | | |
| Acute acquired comitant esotropia | 17 (70.8) | 11 (45.8) | |
| Intermittent esotropia | 3 (12.5) | 6 (25.0) | 0.259 |
| Cyclic esotropia | 1 (4.2) | 5 (20.8) | |
| Partially accommodative esotropia | 2 (8.3) | 1 (4.2) | |
| Non-refractive accommodative esotropia | 1 (4.2) | 1 (4.2)) | |
| **Amblyopia status** | | | |
| Yes | 4 (16.0) | 4 (16.7) | 1.000 |
| No | 21 (84.0) | 20 (83.3) | |
| **Pre–BTXA esotropia at near** | 40.60 ± 7.12 | 44.58 ± 17.69 | 0.303 |
| ≤ 30 PD | 2 (8.0) | 8 (33.3) | |
| > 30–50 PD | 23 (92.0) | 11 (45.8) | **0.002** |
| > 50 PD | 0 (0) | 5 (20.8) | |
| **Pre–BTXA esotropia at distance** | 40.4 ± 6.75 | 45.0 ± 17.69 | 0.232 |
| ≤ 30 PD | 3 (12.0) | 6 (25.0) | |
| > 30–50 PD | 22 (88.0) | 13 (54.2) | **0.016** |
| > 50 PD | 0 (0) | 5 (20.8) | |
| **Diplopia** | | | |
| Yes | 10 (50) | 5 (33.3) | 0.492 |
| No | 10 (50) | 10 (66.7) | |
| **Time from diagnosis to treatment (months)** | 2.58 ± 3.33 | 6.86 ± 9.45 | **0.038** |
| ≥ 2 months | 7 (28.0) | 18 (75.0) | **0.001** |
| < 2 months | 18 (72.0) | 6 (25.0) | |
| **Mean follow-up duration (months)** | 15.49 ± 7.86 | 30.86 ± 19.12 | **0.001** |
| **Anaesthesia** | | | |
| General anaesthesia | 18 (72.0) | 20 (83.3) | 0.496 |
| Topical anaesthesia | 7 (28.0) | 4 (16.7) | |
| **1-month-post-BTXA alignment** | | | |
| Within 10 PD | 10 (50) | 3 (60) | 1.000 |
| Overcorrection | 10 (50) | 2 (40) | |

*Significant < 0.05.

*BTXA* botulinum toxin type A, *PD* prism dioptres, *SD* standard deviation.

revealed a 60.7% success rate in acute acquired comitant esotropia. However, they had a lower median esodeviation and a smaller study population (35 PD, n = 16) compared to that of this study. Additionally, Yu et al. reported a success rate of 68.68% over a 2-year follow-up for acute acquired comitant esotropia and found that the number of hours spent on near work per day was a significant risk factor for esotropia recurrence (Hazard ratio: 1.29, 95% CI = 1.00–1.67) [19]. Another study by Tejedor et al. reported a similar success rate of 52.9% with single BTXA, identifying higher hypermetropia, less

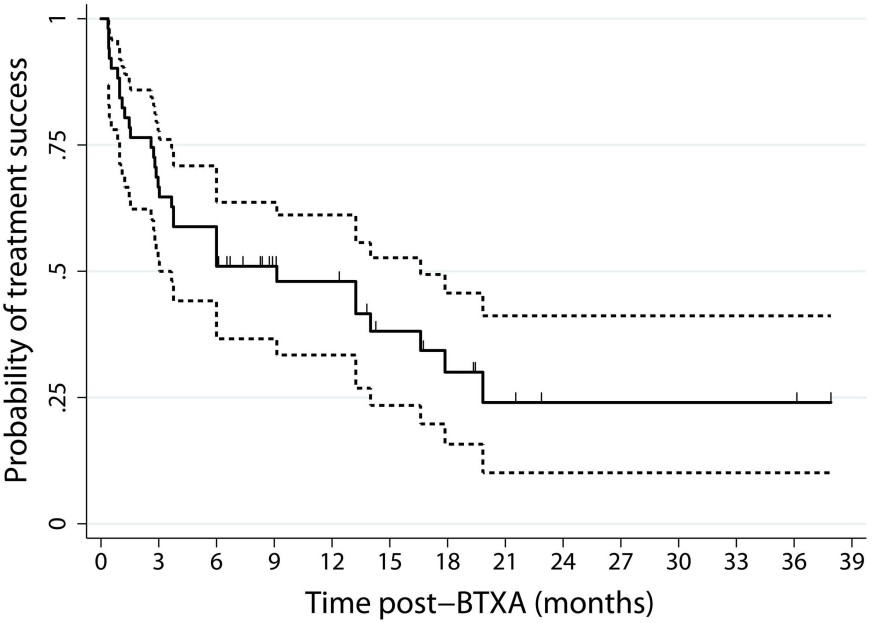

**Fig 1. Kaplan–Meier analysis estimated the probability of treatment success for 49 patients as 51%, 48%, 28%, and 21% at 6-, 12-, 18-, and 24-months intervals, respectively.**

severe amblyopia, and a smaller angle of esotropia, without specifying the exact angular deviations, as predictors of satisfactory outcomes [15]. In another study, Gama et al. [20] showed a 60% success rate with a single BTXA injection in non-accommodative esotropia in children (n = 25, mean esodeviation = 27 PD) with long-lasting effects for up to 2 years. Study by Nguyen et al. reported 89% and 72% success rate at 6-month and 36-month follow-up in acute acquired comitant esotropia [18]. In contrast, our study showed a decline in success rate over time, likely due to the larger mean esodeviation in our patients and longer symptom duration before diagnosis. Nevertheless, our study highlighted that esodeviation >30–50 PD was associated with a success rate similar to that reported by Hered et al. [21] suggesting that BTXA is most effective for small-to-moderate angle deviations (<40 PD). This implies that BTXA is a reasonable treatment for moderate angle deviations. A meta-analysis by Song et al. found that BTXA treatment was as effective as muscle surgery for acute acquired comitant esotropia [22]. Therefore, BTXA is an alternative treatment in such cases. Additionally, our study showed a higher number of overcorrections in the success group, although no significant difference was observed between overcorrection at 1-month post-BTXA or alignment within 10 PD. This contrasts with the findings of Niyaz et al., who reported that early overcorrection is a better predictor of success [23].

Moreover, the incidence of BTXA complications in this study was higher than that in previous studies [18,24]. Consistent with prior research, ptosis emerged as the most common complication, whereas subconjunctival haemorrhage was the least common complication. The increased complication rate might be attributed to the higher dosage of BTXA administered in our study. Furthermore, in our study, esotropia recurrence within 1–2 years of follow-up was observed in the successful group. Therefore, a minimum follow-up period of 2 years is mandatory. Consequently, we recommend a follow-up duration of at least 2 years for patients with successful BTXA treatment due to the risk of esotropia recurrence.

Overall, our study, with a substantial sample size, provides comprehensive insights into the success rate of BTXA as a treatment for childhood-acquired comitant esotropia. Additionally, the analysis of factors influencing successful outcome holds good validity, given the inclusion of predictive factors in multivariate analysis models. This underscores the potential benefits of BTXA as an alternative treatment for acquired esotropia. However, a notable limitation of this study was

**Table 3. Logistic regression analysis predicting factors of successful outcomes.**

| Characteristics | Univariate | | | Multivariate | | |
|---|---|---|---|---|---|---|
| | OR (95% CI) | p value (wald-test) | p value* (LR-test) | OR (95% CI) | p value (wald-test) | p value* (LR-test) |
| **Types of acquired esotropia** | | | | ND | | |
| Acute acquired comitant esotropia | 1 (reference) | | 0.295 | | | |
| Intermittent esotropia | 0.43 (0.07, 1.89) | 0.264 | | | | |
| Cyclic esotropia | 0.13 (0.01, 1.26) | 0.078 | | | | |
| Partially accommodative esotropia | 1.29 (0.10, 16.04) | 0.841 | | | | |
| Non-refractive accommodative esotropia | 0.65 (0.04, 11.45) | 0.767 | | | | |
| **Amblyopia status** | | | | ND | | |
| Yes | 1 (reference) | | 0.950 | | | |
| No | 1.05 (0.23, 4.78) | 0.90 | | | | |
| **Pre–BTXA esotropia at near** | | | | | | |
| ≤ 30 PD | 1 (reference) | | **0.007** | 1 (reference) | | **0.008** |
| > 30–50 PD | 8.36 (1.52, 46.15) | 0.015 | | 6.97 (1.10, 44.13) | 0.039 | |
| **Pre–BTXA esotropia at distance** | | | | NS | | |
| ≤ 30 PD | 1 (reference) | | 0.111 | | | |
| > 30–50 PD | 3.38 (0.72, 15.89) | 0.122 | | | | |
| **Diplopia** | | | | ND | | |
| Yes | 1 (reference) | | 0.322 | | | |
| No | 0.50 (0.13, 1.99) | 0.327 | | | | |
| **Time from diagnosis to treatment** | | | | | | |
| ≥ 2 months | 1 (reference) | | **0.001** | 1 (reference) | | **0.001** |
| < 2 months | 7.71 (2.16, 27.50) | 0.002 | | 6.31 (1.51, 26.41) | 0.012 | |
| **Time from symptom to treatment** | | | | NS | | |
| ≥ 3 months | 1 (reference) | | 0.122 | | | |
| < 3 months | 2.81 (0.73, 10.84) | 0.133 | | | | |
| **1-month-post-BTXA alignment** | | | | ND | | |
| Within 10 PD | 1 (reference) | 0.690 | 0.688 | | | |
| Overcorrection | 1.50 (0.20, 10.99) | | | | | |

*Significant < 0.05.

*CI* confidence interval, *OR* odd ratio, *ND* not done, *NS* not significant, *BTXA* botulinum toxin type A, *PD* prism dioptres.

its retrospective, single-arm design, relatively small subgroup sizes for certain esotropia types, and variable follow-up durations. Our use of a horizontal deviation within 10 PD as the criterion for treatment success post-BTXA is consistent with previous studies and is generally associated with functional binocular vision. Nevertheless, we acknowledge that diplopia may still occur within this range, and our limited binocularity data at 6 months prevent definitive conclusions. This highlights the importance of incorporating both motor and sensory outcomes when defining treatment success. In addition to the identified predictive factors, lifestyle-related visual demands, such as prolonged near work and smartphone use, may influence both initial treatment outcomes and long-term stability. Previous reports have linked excessive screen use with acute acquired comitant esotropia and higher recurrence rates following BTXA or surgery. Although such data were not available in our cohort, these factors warrant further investigation. Therefore, future prospective, multicenter randomized controlled trials with standardized follow-up protocols are recommended to establish more robust evidence supporting the efficacy and safety of BTXA in childhood-acquired comitant esotropia. Nevertheless, the clinical implications of our study suggest that BTXA is an alternative treatment for childhood-acquired esotropia. While our observation

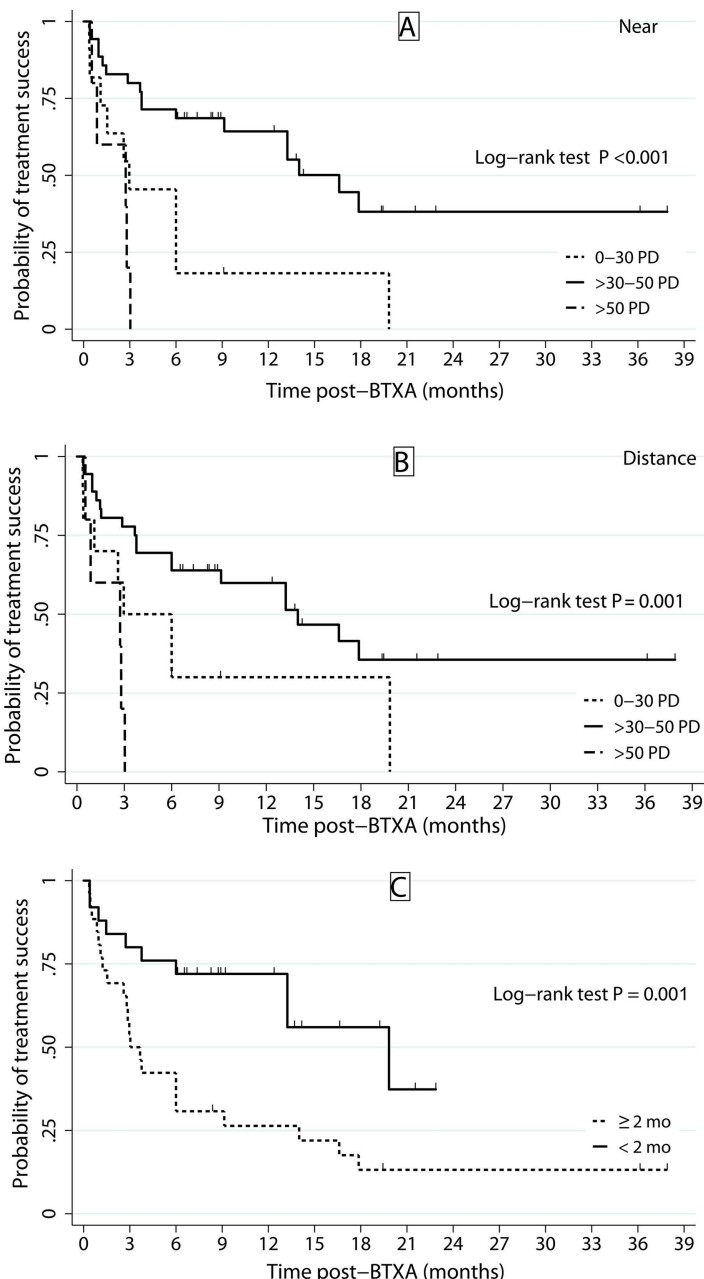

**Fig 2. Kaplan–Meier analysis estimated the probability of treatment success categorised by the range of pre-BTXA esodeviation at near (a) and distance (b), as well as time from diagnosis to treatment (c).**

that smaller-to-moderate angle deviations respond more favourably to BTXA is consistent with prior reports, our multivariate analysis further refined the predictive range to >30–50 PD for optimal outcomes. Additionally, early intervention within 2 months of diagnosis emerged as an independent predictor of success, reinforcing the value of prompt referral and treatment. Particularly, BTXA application potentially reduces procedure time and allows for office-based administration, eliminating anaesthetic risks and the need for hospitalisation. Moreover, BTXA offers the potential to shorten general anaesthesia duration and reduce treatment and hospitalisation costs in younger children requiring general anaesthesia.

## Conclusion

At 6 months, BTXA demonstrated a success rate of 51%. Pre-BTXA esodeviation and the time from diagnosis to treatment were identified as predictive factors for successful outcomes. Therefore, BTXA may be alternative treatment for childhood-acquired comitant esotropia for patients who present early and with an angle of deviation >30–50 PD.

## Acknowledgments

We are grateful to the research consultation department, Ms. Parichat Damthongsuk and Ms. Sujinda Damthong for their suggestions and assistance.

## Author contributions

**Conceptualization:** Narisa Rattanalert, Supaporn Tengtrisorn, Penny Singha.

**Data curation:** Teechaya Nonboonyawat.

**Formal analysis:** Narisa Rattanalert, Teechaya Nonboonyawat, Orapan Aryasit, Juthamat Witthayaweerasak, Manupol Tangthongkum.

**Funding acquisition:** Narisa Rattanalert.

**Methodology:** Narisa Rattanalert.

**Project administration:** Narisa Rattanalert.

**Supervision:** Narisa Rattanalert, Supaporn Tengtrisorn, Penny Singha, Orapan Aryasit, Juthamat Witthayaweerasak.

**Validation:** Narisa Rattanalert.

**Visualization:** Manupol Tangthongkum.

**Writing – original draft:** Teechaya Nonboonyawat.

**Writing – review & editing:** Narisa Rattanalert, Teechaya Nonboonyawat, Supaporn Tengtrisorn, Penny Singha, Orapan Aryasit, Juthamat Witthayaweerasak, Manupol Tangthongkum.

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
