## [Decision Letter · Decision Letter 0]

1 Aug 2025

PONE-D-25-34080Botulinum toxin type A success rates and predictive factors for favourable outcomes in paediatric patients with acquired esotropiaPLOS ONE

Dear Dr. Rattanalert,

Thank you for submitting your manuscript to PLOS ONE. After careful consideration, we feel that it has merit but does not fully meet PLOS ONE’s publication criteria as it currently stands. Therefore, we invite you to submit a revised version of the manuscript that addresses the points raised during the review process.

We look forward to receiving your revised manuscript.

Kind regards,

Neelam Pawar

Academic Editor

PLOS ONE

Journal Requirements:

Additional Editor Comments:

Emphasize on recurrence , post op diplopia and relapse cases.

Reviewers' comments:

Reviewer's Responses to Questions

**Comments to the Author**

1. Is the manuscript technically sound, and do the data support the conclusions?

Reviewer #1: Yes

Reviewer #2: Yes

2. Has the statistical analysis been performed appropriately and rigorously? 

Reviewer #1: Yes

Reviewer #2: Yes

3. Have the authors made all data underlying the findings in their manuscript fully available?

Reviewer #1: Yes

Reviewer #2: Yes

4. Is the manuscript presented in an intelligible fashion and written in standard English?

Reviewer #1: Yes

Reviewer #2: Yes

5. Review Comments to the Author

Reviewer #1: Dear Authors, thank you for your article.

Comments: Why did you considered 10 PD as a success result ? Wouldn't be more accurate to cover the fusional range considering the aspect that one of the stability factors after treatment is binocularity recovery ? At 10 PD ET diplopia can be still present (half of you cohort had pre-toxin diplopia).

I think your data would be more accurate keeping in your statistic only binocular injection.

In my opinion subsequent interventions are not relevant considering the study purpose.

It is possible that, stability after treatment to be influenced by other factors as lifestyle. I our days we confront with Acute Esotropia related to screen overuse, especially smartphones, re-occurrence of ET in not unusual after BT or surgery so, some other factors can be involved in the rate success at two years after injection.

Reviewer #2: I read this article titled as "Botulinum toxin type A success rates and predictive factors for favourable outcomes in paediatric patients with acquired esotropia" with a great interest

1. Congratulations to authors for well written manuscript

2. Line 133-35 It would have been better to mention the recurrence rate ? 7 patients had recurrence

3. Current study brings out the fact that there is 51% success rate with BTXA treatment, with a trend of a decline in success over time. Interestingly only one-fourth of the patients sustained optimal outcomes beyond 2 years.

4.This study does not come up with new knowledge apart from most effective for small-to-moderate angle deviations which is very much known looking at the current literature.

5.Overall, the analysis of factors using univariate model underscores the potential the benefits of BTXA as an alternative treatment for acquired esotropia, and of course this possibly due to the retrospective nature of the study and overall small sample size representing each group of esotropia and less duration of follow up.

6. PLOS authors have the option to publish the peer review history of their article (what does this mean? ). If published, this will include your full peer review and any attached files.

**Do you want your identity to be public for this peer review?** For information about this choice, including consent withdrawal, please see our Privacy Policy .

Reviewer #1: **Yes: ** Daniela Eleonora Cioplean

Reviewer #2: **Yes: ** Vivek Warkad

---

## [Author Response · Author response to Decision Letter 1]

14 Aug 2025

all comment was response. response to reviewers file was attached.

New N = 49

---

## [Decision Letter · Decision Letter 1]

1 Sep 2025

Botulinum toxin type A success rates and predictive factors for favourable outcomes in paediatric patients with acquired esotropia

PONE-D-25-34080R1

Dear Dr. Narisa Rattanalert

We’re pleased to inform you that yo,ur manuscript has been judged scientifically suitable for publication and will be formally accepted for publication once it meets all outstanding technical requirements.

Kind regards,

Neelam Pawar

Academic Editor

PLOS ONE

Additional Editor Comments (optional):

Reviewer #1:

Reviewers' comments:

Reviewer's Responses to Questions

**Comments to the Author**

1. If the authors have adequately addressed your comments raised in a previous round of review and you feel that this manuscript is now acceptable for publication, you may indicate that here to bypass the “Comments to the Author” section, enter your conflict of interest statement in the “Confidential to Editor” section, and submit your "Accept" recommendation.

Reviewer #1: All comments have been addressed

2. Is the manuscript technically sound, and do the data support the conclusions?

Reviewer #1: Yes

3. Has the statistical analysis been performed appropriately and rigorously? 

Reviewer #1: Yes

4. Have the authors made all data underlying the findings in their manuscript fully available?

Reviewer #1: Yes

5. Is the manuscript presented in an intelligible fashion and written in standard English?

Reviewer #1: Yes

6. Review Comments to the Author

Reviewer #1: (No Response)

7. PLOS authors have the option to publish the peer review history of their article (what does this mean? ). If published, this will include your full peer review and any attached files.

**Do you want your identity to be public for this peer review?** For information about this choice, including consent withdrawal, please see our Privacy Policy .

Reviewer #1: **Yes: ** Daniela Eleonora Cioplean

---

## [Editor Report · Acceptance letter]

PONE-D-25-34080R1

PLOS ONE

Dear Dr. Rattanalert,

I'm pleased to inform you that your manuscript has been deemed suitable for publication in PLOS ONE. Congratulations! Your manuscript is now being handed over to our production team.

Kind regards,

on behalf of

Dr. Neelam Pawar

Academic Editor

PLOS ONE